# Cysteamine Suppresses Cancer Cell Invasion and Migration in Glioblastoma through Inhibition of Matrix Metalloproteinase Activity

**DOI:** 10.3390/cancers16112029

**Published:** 2024-05-27

**Authors:** Jinkyu Jung, Orieta Celiku, Benjamin I. Rubin, Mark R. Gilbert

**Affiliations:** 1Neuro-Oncology Branch, Center for Cancer Research, National Cancer Institute, National Institutes of Health, Bethesda, MD 20892, USA; orieta.celiku@nih.gov; 2Department of Ophthalmology, Suburban Hopkins-Hospital, Bethesda, MD 20814, USA; brubin3@mac.com

**Keywords:** cysteamine, matrix metalloproteinase (MMP), glioblastoma (GBM), brain cancer, invasion, migration

## Abstract

**Simple Summary:**

Cysteamine, an endogenously synthesized agent in mammalian cells and FDA-approved for cystinosis, shows promise as a safe anti-cancer agent by inhibiting tumor cell invasion and metastasis. Using computational analyses and multiple glioblastoma cell lines, we demonstrate the in vitro drug efficacy and molecular mechanism by which cysteamine inhibits tumor invasion and migration through targeting matrix metalloproteinase activity at micromolar concentrations. Our findings suggest that cysteamine could represent a novel therapeutic option for the treatment of glioblastoma patients.

**Abstract:**

Glioblastoma (GBM) cells are highly invasive, infiltrating the surrounding normal brain tissue, thereby limiting the efficacy of surgical resection and focal radiotherapy. Cysteamine, a small aminothiol molecule that is orally bioavailable and approved for cystinosis, has potential as a cancer treatment by inhibiting tumor cell invasion and metastasis. Here we demonstrate that these potential therapeutic effects of cysteamine are likely due to the inhibition of matrix metalloproteinases (MMPs) in GBM. In vitro assays confirmed that micromolar concentrations of cysteamine were not cytotoxic, enabling the interrogation of the cellular effects without confounding tumor cell loss. Cysteamine’s inhibition of MMP activity, especially the targeting of MMP2, MMP9, and MMP14, was observed at micromolar concentrations, suggesting the mechanism of action in suppressing invasion and cell migration is by inhibition of these MMPs. These findings suggest that achievable micromolar concentrations of cysteamine effectively inhibit cancer cell invasion and migration in GBM, supporting the potential for use as an adjunct cancer treatment.

## 1. Introduction

Glioblastoma (GBM), a grade IV astrocytoma, is an aggressively invasive brain tumor and is the most common malignant brain tumor in adults. Although GBM rarely metastasizes outside the central nervous system, it characteristically invades brain tissue, preventing cure by surgical excision and limiting the efficacy of local radiotherapy [1]. Although the molecular mechanisms through which this tumor becomes highly invasive are multifactorial, including the overexpression of cell surface peptide such as integrins, the tumor expression of matrix metalloproteinases (MMPs) is an important component of the pathogenesis of invasion [2]. 

MMPs are enzymes involved in collagen remodeling and the wound healing process, with the same mechanism utilized by tumor invasion, metastasis, and angiogenesis [3]. Considerable evidence, research, and publications have indicated that MMPs, including MMP-2 and MMP-9, are actively involved in the spread of malignant cells in tumors by increasing cancer-cell growth, migration, invasion, and angiogenesis [4]. The literature has documented many examples in which MMP’s were associated with the spread of metastasis in GBM [5,6]. 

MMPs have been subtyped and categorized so as to function and be associated with many types of pathology. MMPs are categorized as Zinc dependent, and the MMP’s involved in tumor cell spread have a critical di-sulfide bond. The disruption of the extracellular collagen matrix allows for tumor cell invasion and local contiguous spread [6]. MMP’s are expressed during tumor development, breaking down the tissue barriers to migration and invasion by the tumor cells [7]. In support of the putative role of MMPs in invasion, brain tumors expressing higher levels of MMP2 are frequently associated with higher degrees of invasion, metastasis, and angiogenesis [8]. 

Many MMP inhibitors of varying specificity have been investigated for their anti-cancer effects. Some MMP inhibitors successfully suppress tumor growth and metastases in animal models. However, in clinical trials, the use of MMP inhibitors that are effective in impeding tumor growth and metastases in animal models have produced intolerable adverse effects such as severe joint and muscle pain [9]. Conversely the lower dosages needed to avoid these incapacitating side effects, have not been effective in suppressing tumor spread and invasive metastases. Accordingly, it is important to identify MMP inhibitors that cause few or no side effects [10]. 

In the present study, we examined the anti-invasive effects of cysteamine in several preclinical models of GBM. Cysteamine (HS-CH_2_-CH_2_-NH_2_) is a small, simple sulfhydryl compound that can cross cell membranes due to its small size. Cysteamine is orally bioavailable and has a low toxicity profile and is currently used for the treatment of cystinosis, a cystine storage disorder [11]. The first evidence regarding the therapeutic effect of cysteamine on cystinosis dates to the 1950s. Cysteamine was first approved as a drug for cystinosis in the US in 1994. Extensive data on the pharmacokinetics, toxicity, and LD 50 of cysteamine are available [12]. Our research group has been able to demonstrate that cysteamine can reduce a critical disulfide bond in MMP enzymes, unrelated to its known and documented mechanism in cystinosis, thereby unfolding and inactivating the enzyme and rendering it ineffective. We theorized that the small size of the cysteamine molecule would facilitate penetration through the blood–brain barrier, thereby allowing access to the essential disulfide bond of MMPs in the extracellular matrix of GBMs. 

## 2. Results

### 2.1. MMP2/9 and MMP14 Are Overexpressed in the Microvascular Region of Human GBM Tumors, and Their Expression Levels Are Correlated with Patient Survival and Tumor Grade

MMPs play a crucial role in remodeling the extracellular matrix (ECM) within the tumor microenvironment [13]. The dynamic changes in the ECM influence various cellular processes such as adhesion, migration, and signaling [14]. Cysteamine, as a potential treatment, has the capacity to modulate ECM dynamics by influencing MMP activity [3]. Understanding the interaction between cysteamine and MMPs in the tumor microenvironment is essential for comprehending its impact on the structural and functional aspects of the ECM. However, the maintenance of MMP gene expression in GBM is not fully understood.

To explore MMP expression levels in the tumor microenvironment of GBM, we analyzed gene expression data from the IVY GAP (IVY Glioblastoma Atlas Project) dataset. Many MMP genes are overexpressed in the microvascular proliferation (MVP) region of cellular tumors (CT), with MMP2, MMP9, MMP11, MMP14, MMP15, MMP25, and MMP28 specifically present in this region. This suggests that certain MMPs play a distinct role in the tumor microenvironment of GBM (Figure 1A,B).

To assess the clinical relevance of MMP mRNA expression, we analyzed glioma patient samples from The Cancer Genome Atlas (TCGA). First, patients were grouped in an unsupervised manner based on how similar the expression of MMPs were in their tumors. This unsupervised clustering identified two MMP molecular subtypes, which we denoted as CC1 and CC2. The MMP molecular subtypes were associated with significantly different overall survival (OS). The CC2 subtype, associated with worse OS, exhibited higher expression levels of MMPs compared to the CC1 subtype with better OS (Figure 1C,D). This analysis prompted a focus on specific MMPs more distinctly expressed in CC2, such as MMP2, MMP9, MMP11, MMP14, and MMP25, suggesting their potential role in tumor invasion and their clinical relevance as potential therapeutic targets.

### 2.2. No Cytotoxicity Effect Is Observed in Micromolar Concentrations of Cysteamine Treatment across Multiple GBM Cells

Cysteamine, known for its use in treating cystinosis, has a well-established safety profile, providing a basis for its clinical use. The typical oral dosage for cysteamine bitartrate in cystinosis treatment ranges from 1.3 to 1.95 g/m^2^/day, divided into multiple doses [14]. However, the potential cytotoxicity of cysteamine on in vitro experimental models, including brain cancer, has not been fully explored. In our investigation, we examined whether micromolar doses of cysteamine were effective for GBM cell viability. In vitro cell viability assays with cysteamine treatment in a dose-dependent manner were performed on 11 GBM cell lines, and normal human astrocytes (NHA), (Figure 2A,B). Concentrations of cysteamine less than 0.2 mM did not significantly decrease cell viability or tumor cell growth in GBM cell lines (Figure 2A). The IC50 result indicated that GBM cells responded only to millimolar concentrations of cysteamine, suggesting that lower concentrations may be applied without causing significant cytotoxicity (Figure 2C). Molecular assays for cell death-related markers through western blots further confirmed that micromolar concentrations (50, 100, and 500 μM) of cysteamine did not induce cell death markers, including cleaved caspase 3 and cleaved PARP 1, along with LC3B activation as an autophagy marker (Figure 2D). Overall, these results suggest that micromolar concentrations of cysteamine caused no cytotoxicity through cell death signaling pathways.

### 2.3. Cysteamine Inhibits MMP Activity in GBM Cells, Specifically Targeting MMP2, MMP9, and MMP14

To determine if the action mechanism of cysteamine treatment is dependent on MMP activity, we treated U251 and LN229 GBM cells with cysteamine in a dose-dependent manner, followed by assessing the total enzymatic and specific MMP activity levels. Cysteamine treatment decreased MMP activity in a dose-dependent manner in these representative GBM cells (Figure 3A,B). The IC50 for total MMP activity, observed at a ~0.1 mM concentration, suggested that cysteamine may effectively suppress invasion and cell migration through the inhibition of MMP activity at micromolar concentrations. Specific MMP ELISA assays for MMP2, MMP9, and MMP14 revealed that cysteamine specifically decreased levels of these MMPs, indicating that the molecular mechanism of cysteamine treatment is dependent on the activity of MMP2, MMP9, and MMP14 (Figure 3C–E). Transglutaminase 2 (TGM2) plays a crucial role in the epithelial–mesenchymal transition and TRAIL resistance; its increased level contributes invasion and migration in cancer cells [15]. We found that mRNA expression levels of TGM2 and MMP2/MMP9/MMP14 were positively correlated in the TCGA GBM patient database, suggesting that TGM2 may be involved in MMP pathways in GBM (Figure 3F). To further investigate the molecular signaling pathways of MMP inhibition by cysteamine treatment, we conducted western blots for transglutaminase 2 (TGM2) and extracellular matrix proteins such as N-cadherin (an invasion promoter) and E-cadherin (a tumor suppressor). The cysteamine treatment downregulated TGM2 expression and N-cadherin but upregulated E-cadherin, suggesting that the inhibition of TGM2 by cysteamine led to decreased MMP2, MMP9 and MMP14 activity, followed by a change in the expression of cadherin genes (Figure 3G). These findings demonstrate that micromolar concentrations of cysteamine treatment specifically inhibit MMP activity and TGM2 expression, subsequently decreasing MMP2, MMP9, and MMP14 protein levels in GBM cells.

### 2.4. Cysteamine Inhibits the Invasion and Migration of GBM Cells

To assess the impact of cysteamine treatment on the invasion of GBM cells, we conducted a Matrigel invasion assay in the presence of increasing levels of cysteamine. Six doses of cysteamine, from 0 to 5 mM, selected based on concentrations used in the cell viability test, were applied to GBM cell lines U251 and LN229 for the invasion assay. Cysteamine treatment led to a reduction in the number of invaded cells in both GBM cell lines. Specifically, micromolar concentrations (0.1 and 0.5 mM) of cysteamine, which showed no cytotoxicity, significantly inhibited the invasion of GBM cells (Figure 4A,B). At millimolar concentrations (>1 mM), morphological changes potentially processed by cell death programming in GBM cells and a decrease in invaded cells were observed concurrently, suggesting that higher concentrations of cysteamine might impact GBM cells with cytotoxicity. Furthermore, to investigate the effect of cysteamine treatment on the migration of GBM cells, we conducted a wound healing assay in a dose-dependent manner. A significant decrease in cell migration was observed at and above 0.1 mM of cysteamine in both GBM cell lines, U251, and LN229 (Figure 4C). Taken together, these results suggest that micromolar concentrations of cysteamine treatment are effective in inhibiting cancer cell invasion and migration in GBM cells without causing excessive cytotoxicity.

## 3. Discussion

This study investigates the role of matrix metalloproteinases (MMPs) in GBM cell lines and explored the potential therapeutic effects of cysteamine. Cysteamine, a potential treatment, was explored for its ability to modulate cancer cell invasion and migration by influencing MMP activity in GBM cells. MMPs play a crucial role in remodeling the extracellular matrix (ECM) within the tumor microenvironment, influencing cellular processes such as adhesion, migration, and signaling [16]. Our findings show that specific MMPs including MMP2, MMP9, and MMP14 are overexpressed in the microvascular region of GBM patient tumors. The expression levels of these MMPs correlate with tumor grade and patient survival, suggesting their significance in GBM progression. Analysis of the anatomic gene expression data revealed overexpression of various MMP genes in the microvascular region of GBM tumors. Specific MMPs, including MMP2, MMP9, MMP11, MMP14, MMP15, MMP25, and MMP28, were increased in this region, indicating their distinct roles in the tumor microenvironment and confirming the importance of understanding the effects of cysteamine on specific MMPs in GBM cancer models. In this regard, cysteamine treatment inhibits MMP activity in a dose-dependent manner and decreases MMP2, MMP9, and MMP14 protein levels in GBM cells. Finally, micromolar concentrations of cysteamine are effective in suppressing GBM cell invasion and migration. Taken together, these findings provide strong support for the role of MMPs in tissue invasiveness and patient prognosis in GBM where cysteamine inhibits MMP activity and decreases MMP protein levels in GBM cells and decreases GBM cell invasiveness and migration in cultures. Additionally, these findings provide a rationale for a possible clinically therapeutic role for cysteamine in delaying and decreasing GBM invasiveness in patients.

The selective inhibition of targeted MMPs remains a challenge in the successful development of MMP inhibitors for clinical trials in cancer treatment due to the heterogeneous expression of MMPs in cancer cells and the crosstalk within the MMP family in signaling pathways [17]. For instance, MMP14 functions by forming complexes with itself on the cell surface and with MMP2 and TIMP2 to activate MMP2 [18]. Therefore, it is crucial to develop specific MMP inhibitors capable of targeting specific types of cancers. Six inhibitors have been identified for specifically targeting MMP14 along with MMP2 and MMP9 [17], suggesting that our findings can be applied to a broader range of disorders. Thus, this study provides insights into the overexpression of specific MMPs in GBM and highlights the potential for a therapeutic role of cysteamine in inhibiting MMP activity, invasion, and migration of GBM cells, using clinically achievable micromolar concentrations to avoid cytotoxicity.

Prior research in our facility demonstrated that cysteamine suppresses metastasis in pancreatic cancer. That study showed a decrease in movement of the cancer cells in a laboratory model as well as a decrease in pancreatic cancer metastasis in an orthotopic animal model using an immunodeficient mouse. With GBM, we were similarly able to show a decrease in the movement of the cancer cells in a laboratory model. In vivo testing using mouse models was not pursued with GBM because currently available mouse models are not indicative or reflective of GBM in humans due to the rapid growth of these tumors and minimal degree of local tumor cell invasion in these models. The success of the studies using cell line models for both pancreatic cancer and GBM supports the role of cysteamine in decreasing the metastatic spread of GBM, notwithstanding the fact that an animal model was not utilized. 

Traditional systemic therapies for GBM including cytotoxic chemotherapy and signal transduction modulators have only modest efficacy, probably due to both disease heterogeneity and the limited delivery of therapeutic reagents through the blood–brain barrier. The modulation of tumor invasiveness may help augment treatment, potentially limiting the extent of brain invasion to distant regions where drug delivery is more difficult. Our findings show that cysteamine, as a potential treatment for GBM, may offer benefits in overcoming some of these limitations: (1) Cysteamine has been shown to influence the activity of matrix metalloproteinases (MMPs), enzymes involved in extracellular matrix (ECM) remodeling and cancer cell invasion. By inhibiting MMPs, cysteamine may help limit the invasive potential of GBM cells. (2) Cysteamine has a well-established safety profile from its use in treating cystinosis. This existing safety data provides a foundation for considering its application in cancer therapy. (3) Combining cysteamine with other treatment modalities, such as surgery, radiation, or chemotherapy, may enhance overall therapeutic efficacy. The ability of cysteamine to modulate ECM dynamics and inhibit MMP activity may complement existing treatments by targeting the invasive nature of GBM. (4) Cysteamine, at micromolar concentrations, has shown effectiveness against invasion by GBM cells without inducing significant cytotoxicity. This suggests that it may have a very wide therapeutic window so that, at effective concentrations, there will be minimal toxicity to the brain and other organs. In conclusion, cysteamine may have clinical utility in altering GBM invasion, thereby enhancing other therapies, although this requires formal testing in clinical trials. 

## 4. Materials and Methods

### 4.1. Reagent, Human Glioma Cell Culture, and Mycoplasma Testing

Cysteamine hydrochloride was purchased from Sigma-Aldrich (Burlington, MA, USA) (cat #M6500) and dissolved in distilled water. Human patient-derived GBM stem cells were obtained after surgery from patients at the multiple research institutes (University of Pittsburgh, the University of Virginia, University of Florida, and NCI/NIH). GSC XO-8 and XO-9 were isolated from the University of Virginia. GSC L0, L1, CA1, CA2, and CA4 were from the University of Florida. GSC 827 and 923 were isolated from the Neuro-Oncology Branch, NCI, NIH. GSCs were maintained as tumorsphere cultures in neurobasal media (cat #A2477501, Gibco (Waltham, MA, USA)) supplemented with B27/N2 supplement (minus vitamin A, cat #12587010, ThermoFisher (Waltham, MA, USA) and cat # A1370701, ThermoFisher), 20 ng/mL EGF, 20 ng/mL bFGF, and penicillin/streptomycin, as described in a standard protocol. Normal human astrocytes (NHA, cat # CC-2565, Lonza (Basel, Switzerland)) were cultured according to the manufacturer’s protocol. GBM cell lines (U251 and LN229) were donated from laboratories of our research institute and cultured in the following DMEM media (cat #11965-092, Gibco), including 10% fetal bovine serum and penicillin/streptomycin. We used low passage cell lines (5–15 cycles). All cell lines are routinely tested for mycoplasma contamination at the NCI Frederick core facility and were negative.

### 4.2. Cell Viability Assay

Tumor viability was performed using CellTiter-Glo reagent (cat # G7570, Promega (Madison, WI, USA)). The plate, including the cells, were incubated at room temperature for 30–45 min for equilibration. CellTiter-Glo reagent was added to cell culture medium with an equal volume to the medium. The plate was incubated at room temperature for 10 min to stabilize luminescent signal. A multiple-plate reader was used for detection of the signal. GraphPad Prism (Version 10.2.3) and Excel software were used for further analysis.

### 4.3. Western Blotting

Proteins were extracted using CelLyticTM M (for cell; cat #2978, Sigma) cell lysis buffer supplemented with protease inhibitor cocktail and phosphatase inhibitors. Protein levels were determined with western blotting using conventional protocols. Proteins were detected using specific primary antibodies from cleaved caspase 3 (cat #9664, cell signaling), cleaved Parp1 (Cat# 5625, cell signaling), LC3B (cat #3868, cell signaling), and GAPDH (cat #5174, cell signaling); and subsequently with the appropriate horseradish peroxidase (HRP)-conjugated secondary antibodies (cat #7074—anti-rabbit, cat #7076—anti-mouse, cell signaling). Immobilon western chemiluminescent HRP substrate kit (cat #WBKLS0500, EMD Millipore (Burlington, MA, USA)) was used to visualize protein bands. For protein bands quantification from western blotting, ImageJ software (https://imagej.net/ij/, accessed on 25 January 2024, National Institutes of Health (Bethesda, Rockville, MD, USA)) was used for converting band intensity to a numerable value. The fold difference of interest was calculated compared to a control, normalized by GAPDH levels.

### 4.4. MMPs Activity Measurement

For overall activity of MMPs, fluorescence-based MMP activity was determined using the manufacturer’s protocol (Abcam (Cambridge, UK), #ab112146). MMP green substrate and 40 μg of cell lysate proteins were added to a 96-well plate, and fluorescence was measured at an excitation/emission of 490/525 nm. For MMP2, MMP9, and MMP14, a human-specific MMP2/MMP9/MMP14 solid-phase sandwich ELISA (enzyme-linked immunosorbent assay), designed to measure the amount of the target bound between a matched antibody pair, was performed following the manufacturer’s instructions (Thermo Fisher Scientific, KHC3081/BMS2016-2). Cell lysates were collected and centrifuged at 12,000 rpm for 10 min, and, after centrifugation, the supernatant was collected. Forty micrograms of proteins for each sample was used for these assays, which were carried out according to the manufacturer’s protocols.

### 4.5. Matrigel Invasion Assay

Invasion capacity was assessed using Corning BioCoat Matrigel Invasion Chambers with BD Matrigel Matrix (Thermo Fisher Scientific). Cells were treated with cysteamine in a dose-dependent manner before seeding into the Matrigel invasion chamber for 24 h. Pre-coated membranes (8 μm pore size, Matrigel 100 μg/cm^2^) were rehydrated and seeded with 2 × 10^5^ cells in 2 mL of DMEM media, in triplicates into the upper part of each chamber. The lower compartment was filled with 2 mL of DMEM media. After a 24 h incubation, the non-invaded cells on the upper surface of the membrane were removed with a cotton swab. Invaded cells on the lower surface of the filter were fixed and stained with a 0.5% crystal violet solution for 20 min, and invasive cells were quantified by counting. 

### 4.6. Wound Healing Assay

For the migration assay, a wound healing assay was performed. Cells were seeded onto 12-well plates at a density of 2 × 10^5^ cells. When the cells reached 95–100% confluence, a scratch was created using a 200 μL pipette tip, and the cells were then treated with cysteamine. After 24 h, the created wounds were observed every 24 h and photographed using an EVOS cell imaging system (EVOS XL core cell imaging system, Thermo Fisher Scientific).

### 4.7. In Silico Analysis

For computational analysis, normalized RNASeq expression data were downloaded from TCGA as R MultiAssayExperiment objects, and the IVY GAP database was used (https://glioblastoma.alleninstitute.org, database was accessed on 25 January 2024). The total 662 glioma samples (516 Lower Grade Gliomas and 156 GBMs) of the TCGA database were utilized to analyze the MMP gene expression in glioma patient samples. The IVY GAP (IVY Glioblastoma Atlas Project) dataset was utilized to assess MMP expression levels. All glioma patient samples from TCGA were employed for unsupervised clustering based on the expression of MMP genes, which resulted in two MMP molecular subtypes: CC1 and CC2. Kaplan Meier overall survival (OS) analysis was conducted with a log-rank test to assess differences in the overall survival of patients based on CC1 versus CC2 subtype membership.

### 4.8. Statistical Analysis

The significance of the differences was evaluated using the independent Student’s *t*-test. The statistical analyses were performed with GraphPad Prism and Microsoft Excel software; *p*-value significant level represents * for *p*-value < 0.05, ** for *p*-value < 0.01, and *** *p*-value < 0.001, respectively.

## 5. Conclusions

We found that cysteamine treatment at micromolar concentrations specifically targets MMP2, MMP9, and MMP14, leading to the inhibition of tumor invasion and migration in GBM cells overexpressing these genes. These findings suggest that clinically achievable concentrations of cysteamine are effective in inhibiting GBM invasiveness, supporting its potential use as an adjunct cancer treatment.

## Figures and Tables

**Figure 1 cancers-16-02029-f001:**
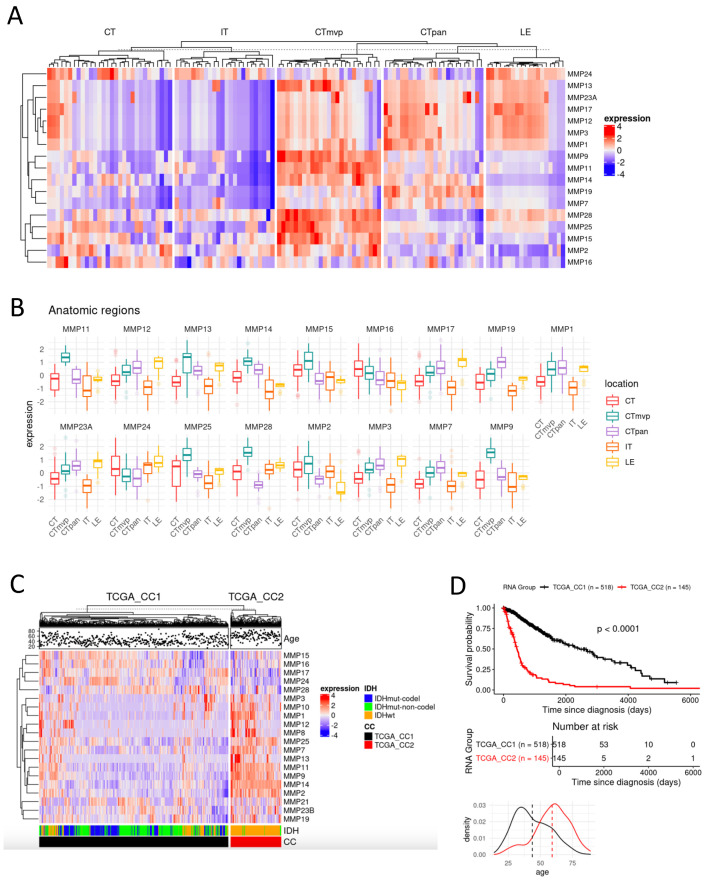
MMP2/9 and MMP14 are overexpressed in the microvascular region in GBM, and their expression level is associated with patient survival and tumor grade. (**A**,**B**) The IVY GAP (IVY Glioblastoma Atlas Project) dataset was utilized to assess MMP expression levels in different anatomic tumor regions, including the leading edge (LE), infiltrating tumor (IT), cellular tumor (CT), pseudopalisading cells around necrosis (PAN), and microvascular proliferation (MVP). (**C**,**D**) All glioma patient samples from TCGA were employed for clustering based on the expression of MMP genes. Unsupervised clustering identified two MMP subtypes, denoted CC1 and CC2. CC2 membership was associated with significantly worse overall survival (OS); the subtype was predominantly composed of IDH wild type samples and was characterized by overexpression of distinct MMPs. The CC1 was associated with significantly better survival; this subtype is enriched in IDH mutant samples but also includes IDH wild type samples.

**Figure 2 cancers-16-02029-f002:**
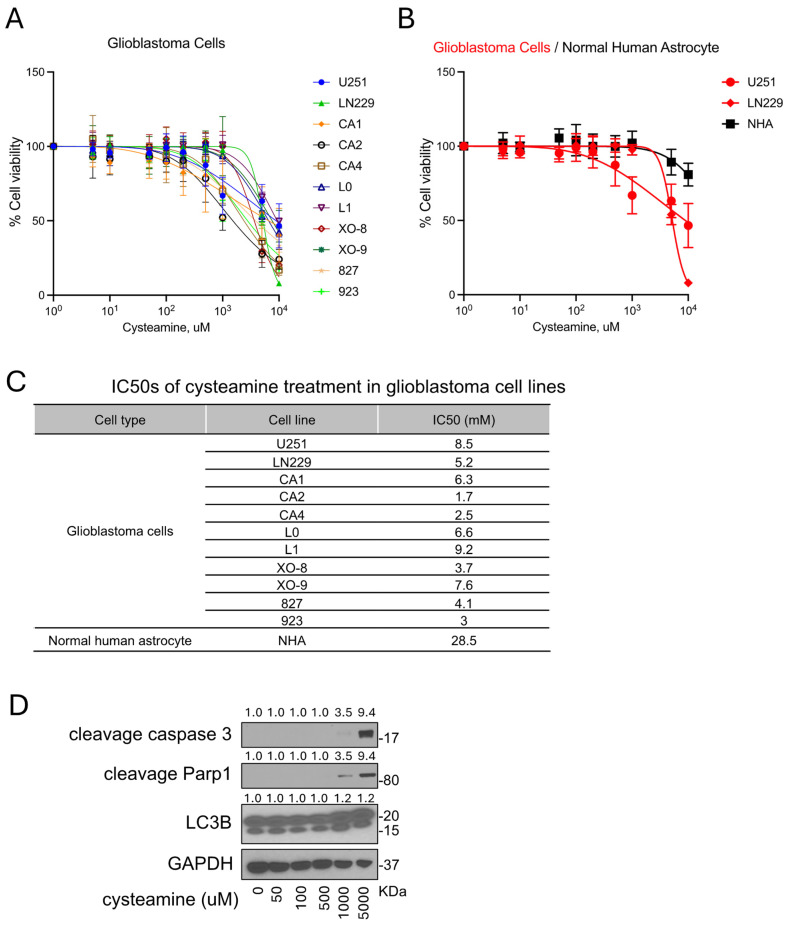
Micromolar concentrations of cysteamine do not affect tumor cell growth and cell death in GBM. (**A**) GBM cell viability was assessed using CellTiter-Glo assay. Briefly, 1 × 10^3^ cells of each cell line were seeded per well in a 96-well plate and incubated overnight for drug treatment. The following day, the cells were treated with multiple concentrations of cysteamine (0, 0.005, 0.01, 0.05, 0.1, 0.2, 0.5, 1, 5, and 10 mM) for 48 h, followed by CellTiter-Glo cell viability assays. (**B**) The cell viability assay was performed in NHA cells, and the result was compared to U251 and LN229. (**C**) The percentages of cell viability and IC50 were calculated using Prism and Excel software. (**D**) U251 cells were incubated with cysteamine in a dose-dependent manner (0, 0.05, 0.1, 0.5, 1, and 5 mM) for 48 h, as indicated, followed by western blotting for cleaved caspase 3, cleaved Parp1, LC3B, and GAPDH. Original western blots are presented in Appendix A.

**Figure 3 cancers-16-02029-f003:**
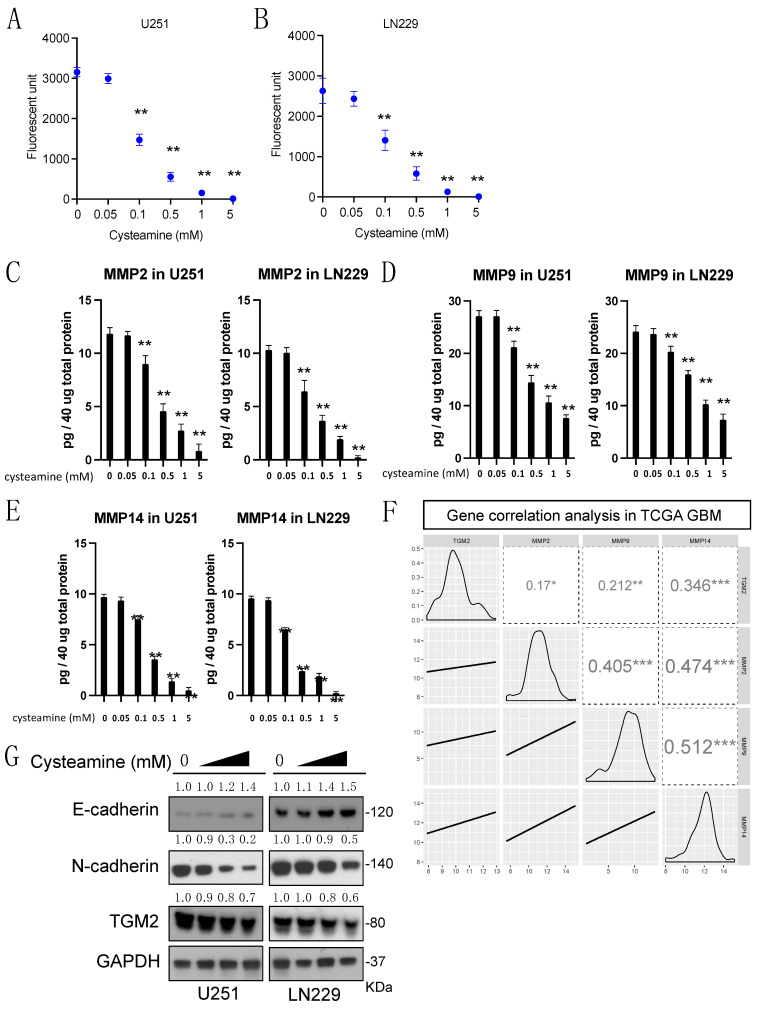
Cysteamine decreases MMP2/9 and MMP14 activity in GBM. (**A**,**B**) U251 and LN229 cells were cultured and treated with cysteamine for 48 h in a dose-dependent manner (0, 0.05, 0.1, 0.5, 1, and 5 mM) to assess changes in invasion and migration. Fluorescence-based MMP activity was determined using the manufacturer’s protocol. (**C**–**E**) A human-specific MMP2/MMP9/MMP14 solid-phase sandwich ELISA (enzyme-linked immunosorbent assay), designed to measure the amount of the target bound between a matched antibody pair, was performed. U251 and LN229 cells, in response to changes in invasion and migration, were cultured and treated with cysteamine for 48 h in a dose-dependent manner (0, 0.05, 0.1, 0.5, 1, and 5 mM). Cell lysates were collected, centrifuged at 12,000 rpm for 10 min, and, after centrifugation, the supernatant was collected. Forty micrograms of proteins for each sample were used for this assay, which was carried out according to the manufacturer’s protocols. (**F**) Correlation between TGM2 and MMPs mRNA expression in all TCGA GBM specimens. R-value for the coefficient of correlation was represented as shown in the figure. The significance of the differences was evaluated using the independent Student’s *t*-test; *p*-value significant level represents * for *p*-value < 0.05, ** for *p*-value < 0.01, and *** *p*-value < 0.001, respectively. (**G**) U251 and LN229 cell lines were incubated with cysteamine in a dose-dependent manner (0, 0.05, 0.1, and 0.5 mM) for 48 h, followed by western blotting for E-cadherin, N-cadherin, TGM2 and GAPDH. Original western blots are presented in Appendix A.

**Figure 4 cancers-16-02029-f004:**
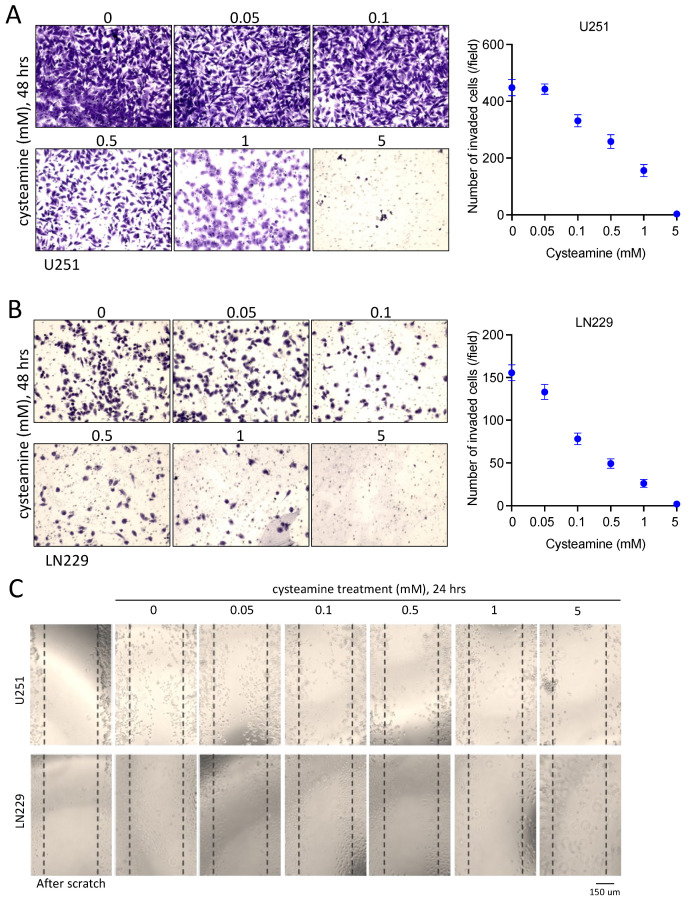
Cysteamine inhibits cancer cell invasion and migration. (**A**,**B**) GBM cell-invasive capacity was assessed using Matrigel invasion assay. Cells were treated with cysteamine in a dose-dependent manner (0, 0.05, 0.1, 0.5, 1, and 5 mM) before seeding into the Matrigel invasion chamber for 24 h, followed by invasion assay as shown in the Section 4. (**C**) GBM cells were seeded onto 12-well plates at a density of 2 × 10^5^ cells. When cells reached 95–100% confluence, a scratch was created using a 200 μL pipette tip, and cells were then treated with cysteamine in a dose-dependent manner (0, 0.05, 0.1, 0.5, 1, and 5 mM). After 24 h, the created wounds were observed every 24 h and photographed using an EVOS cell imaging system.

## Data Availability

Data are available upon request from the corresponding authors.

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
