# Peer review of "Cysteamine Suppresses Cancer Cell Invasion and Migration in Glioblastoma through Inhibition of Matrix Metalloproteinase Activity"

_cancers, 2024, doi:10.3390/cancers16112029_

Round 1

Reviewer 1 Report

Comments and Suggestions for Authors

The manuscript investigates the role of matrix metalloproteinases (MMPs) in glioblastoma (GBM) and explores the potential therapeutic effects of cysteamine. Firstly, the study examines the expression levels of MMPs in GBM tumors and their correlation with patient survival and tumor grade. Using gene expression data from The Cancer Genome Atlas (TCGA) and the IVY Glioblastoma Atlas Project, the study identifies specific overexpressed MMPs, including MMP2, MMP9, and MMP14, are associated with tumor progression and poorer patient outcomes. The study then investigates the effects of cysteamine on GBM cells. Cysteamine is found to specifically target MMP2, MMP9, and MMP14. Furthermore, cysteamine treatment suppresses GBM cell invasion and migration without inducing significant cytotoxicity. Overall, the findings suggest that cysteamine serves as a therapeutic agent for GBM by targeting MMP activity and reducing tumor invasiveness. However, several key questions remain:

1. While the manuscript provides strong evidence from in vitro experiments, there is a lack of in vivo validation of the findings.

2. The authors could assess the efficacy of combining cysteamine with existing GBM treatments.

3. It would be beneficial to explore the underlying molecular mechanisms through which cysteamine inhibits MMP activity and decreases tumor invasiveness.

Reviewer 2 Report

Comments and Suggestions for Authors

The submitted manuscript presents the work on the application of cysteamine to inhibit chosen metalloproteinases. The manuscript is nicely written, clear, concise, convincing. However, it also requires some revisions. My suggestions are listed below.

Title-MMP abbreviation must be explained in the title

Line 46, this sentence should start in a new paragraph

Line 52, a reference is needed here

Line 68, are there any MMPs inhibitors currently on the global market?

Line 70, it should be HS-CH2-CH2-NH2

The Authors haven’t described the origin (manufacturers) of the reactants used for this study, including cysteamine

Figure 2A, due to the use of the same (red) color for almost all of the series, this chart is completely unreadable. The Authors should improve it, either by splitting into more figures or using different colors

Figure 3, those data should undergo statistical analysis to determine if there are significant differences between them

Also, the Authors should discuss the possible mechanisms of MMP inhibition – is cysteamine a competitive/non-competitive inhibitor? What are the possible binding site(s)?

A short “Conclusions” section should be created

At the end of the manuscript, the part describing each author’s individual contribution is missing. This is mandatory in all MDPI journals.

Round 2

Reviewer 2 Report

Comments and Suggestions for Authors

The Authors have significantly improved their work and answered all of my questions. This version can be accepted.